

# Obstetric and perinatal outcomes of dizygotic twin pregnancies resulting from in vitro fertilization versus spontaneous conception: a retrospective study

Hua Chen[1], Ying Wan[2], Haitao Xi[1], Weijue Su[3], Jing Cheng[1], Chunfang Zhu[1], Jieqiang Lv[1], Xinmei Wu[4] and Junzhao Zhao[1]

[1] Reproductive Center, Department of Obstetrics and Gynecology, The 2nd Affliated Hospital and Yuying Children's Hospital of Wenzhou Medical University, Wenzhou, China
[2] Department of Obstetrics and Gynecology, Jiaxing Maternal and Child Health-care Hospital, Jiaxing, China
[3] Wenzhou Medical University, Wenzhou, China
[4] Department of Clinical Laboratory, The 2nd Affliated Hospital & Yuying Children's Hospital of Wenzhou Medical University, Wenzhou, China

## ABSTRACT

This study was designed to to assess perinatal and neonatal outcomes of dizygotic twin pregnancies conceived naturally or by in vitro fertilization (IVF). After strict selection, the study included 470 dizygotic twin pregnancies. There were 249 resulting from IVF treatments and 221 conceiving spontaneously. After adjusting maternal age and primiparity, the results showed that there were no significant differences between the two groups ($P > 0.05$) in terms of maternal antenatal complications and neonatal outcomes. In conclusion, our study does not reveal increased risks for pregnancy-related complications and adverse neonatal outcomes in dizygotic twin pregnancies following IVF treatments. With these fundamental data, this study could provide a reference for perinatal care and clinical assisted reproductive technology (ART) treatment and help to inform infertile parents about the potential risks of IVF treatments.

## INTRODUCTION

The rapid progress and wide application of assisted reproductive technology (ART), especially in vitro fertilization and embryo transfer (IVF-ET), make it possible for infertile couples to conceive a baby successfully. Unlike spontaneous conception, however, to maximize the possibility of obtaining a live birth, two or more embryos are routinely transferred into the uterus (*Vasario et al., 2010*). Consequently, the incidence of multiple gestation rapidly rises (*Farhi et al., 2013*), which in turn raises perinatal and neonatal risks (*Qin et al., 2015*).

In spontaneous-conceived (SC) pregnancies, the incidence of multiple gestation is about 2%, while this rate is up to 40%–50% in in vitro fertilization (IVF)-conceived pregnancies (*Reynolds et al., 2003*). Notably, twin pregnancy accounts for the majority of multiple

Corresponding authors
Xinmei Wu, 205046@wzhealth.com, applewuxinmei@126.com
Junzhao Zhao, z.joyce08@163.com

gestations. Although single embryo transfer has been suggested, the implementation of single embryo transfer may reduce the success rate. Therefore, how to successfully carry it out without compromising the clinical outcomes is still a technological challenge. To date, the strategy of at least two embryos for transfer is routinely performed in most reproductive centers.

Although twin pregnancies are known to be related to higher risks of maternal and fetal complications than singleton pregnancies in natural conception (*Qin et al., 2015*; *Rao, Sairam & Shehata, 2004*), it is not yet clear whether twin pregnancies following IVF treatments are associated with higher risks of obstetric and neonatal conditions when compared with SC twin pregnancies. Several studies have been performed to explore the exact role of IVF-ET in obstetric and neonatal outcomes of twin pregnancies, but the results remain conflicting (*Geisler et al., 2014*; *Helmerhorst et al., 2004*; *McDonald et al., 2005*; *Pourali et al., 2016*; *Qin et al., 2015*; *Sun et al., 2016*; *Vasario et al., 2010*). One of the confounding factors in some studies is that monochorionicity among IVF-conceived twin pregnancies is quite rare, when compared with that in SC twin pregnancies (about 2% and 22%, respectively), and monochorionic pregnancies have more adverse effects on obstetric outcomes (*Penava & Natale, 2004*). As a result, the adverse impact of IVF treatments on the outcomes of IVF-conceived twins may be compensated by the lower incidence of monochorionicity. Accordingly, it is necessary to exclude monozygotic twins and merely consider dizygotic twins. In addition, other factors responsible for the conflicting outcomes can be ascribed to the differences in study populations and the management approaches in twin pregnancies related studies.

Given current evidence on this topic is contradictory, and the results in medical literatures might vary greatly due to study populations, management approaches, and sample size. This study is thus designed to assess perinatal and neonatal outcomes of dizygotic twin pregnancies conceived naturally or by IVF.

## MATERIALS AND METHODS

This retrospective study was performed at the Second Affiliated Hospital of Wenzhou Medical University during the period from July 2015 to April 2016, and was approved by the Ethics Committee of the hospital. Informed consent was not needed for this retrospective investigation. The subjects in this study were mainly from Wenzhou area and its surrounding area, South-East Zhejiang province, China.

Patients who had antenatal care and delivered at Obstetric department of the Second Affiliated Hospital of Wenzhou Medical University during the period from July 2015 to April 2016 were taken into consideration. Only dizygotic twin pregnancies delivered after 28 weeks' gestation were included. In the IVF-conceived group, only those conceived following IVF/ICSI treatments were included, those conceived by other forms of assisted reproduction technology were excluded. Twin gestations obtained after natural abortion or fetal reduction in multiple pregnancies were excluded. After selection, 470 patients were

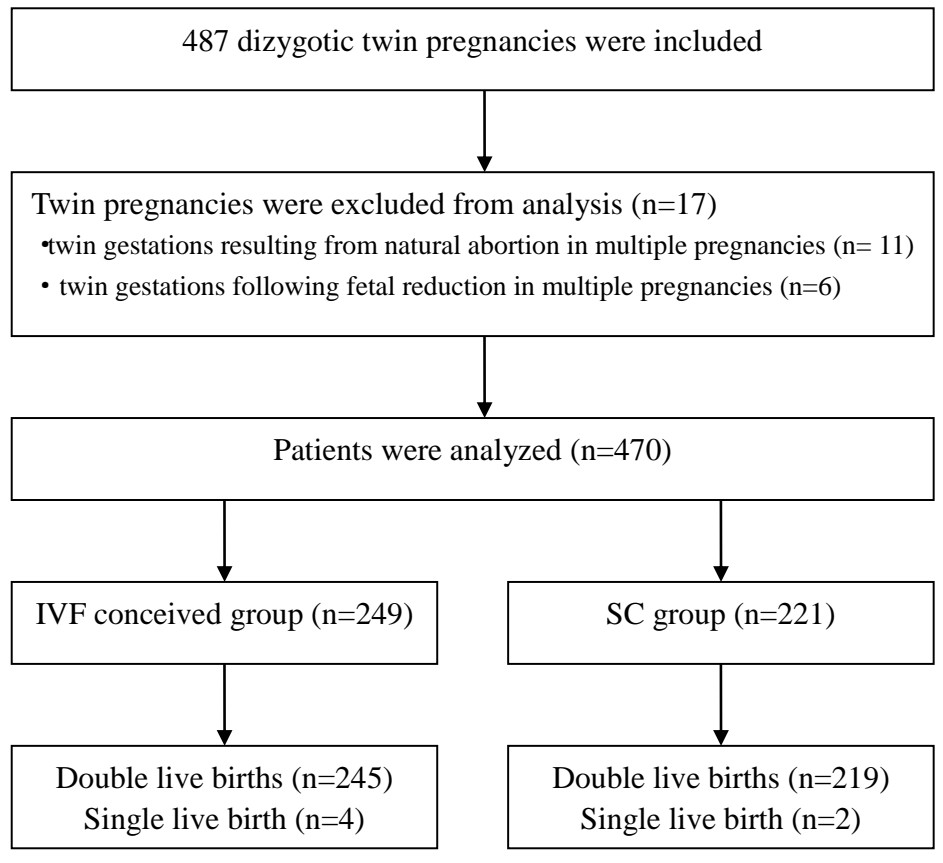

Figure 1. A workflow to show inclusion criteria in the study

**Figure 1**   **A workflow to show inclusion criteria in the study.**

included; 249 dizygotic pregnancies conceived by IVF and 221 conceived spontaneously, as described in Fig. 1.

Data were obtained from medical records, which include maternal age, gestational age, delivery mode, duration of hospitalization, perinatal complications, neonatal birth defects, birthweight, and neonatal intensive care unit (NICU) admission. The medical diagnosis was categorized according to the criteria proposed by *Sun et al. (2016)*.

The diagnosis of dichorionic twin pregnancies was based on ultrasounds between 6 and 10 weeks' gestation by the observation of two gestational sacs on vaginal sonography in the first trimester. Chorionicity was confirmed by examination of the placenta following delivery. Gestational age was calculated from 14 days before the embryo transfer for IVF-conceived women, and from the first day of the last menstrual cycle for SC women.

Length of hospitalization was calculated from the day of admission to the day of discharge after delivery.

Gestational hypertension is the development of new hypertension in a pregnant woman after 20 weeks' gestation without the presence of protein in the urine or other signs of preeclampsia. Hypertension is defined as having a blood pressure greater than 140/90 mmHg. Gestational diabetes mellitus (GDM) was diagnosed by oral 75 g glucose tolerance test (OGTT) at between 24 and 28 weeks' gestation. According to the International association for research on gestational diabetes 2010. The glucose threshold of diagnosing GDM is 5.1 mmol/L for fasting blood test, 10.0 mmol/L at 1 hour after OGTT, 8.5 mmol/L at 2 h after OGTT. Premature rupture of membranes was documented as rupture of membranes prior to the woman going into labor. Maternal anemia was determined as a decrease in whole-blood hemoglobin concentration of more than standard deviations below the mean of an age-matched reference range. Placenta previa was recorded as after 28 weeks of gestation, the placenta attaches to the lower segment of the uterus, and even the lower margin of the placenta reaches or covers the cervical orifice, and its position is lower than the first exposed part of the fetus. Placental acrete is defined as the abnormal adherence to or invasion of the villous tissue into the myometrium secondary to damage to the endometrium-myometrial interface of the uterine wall. Postpartum hemorrhage was defined as vaginal blood loss of ≥500 ml within 24 h after delivery, and when blood loss during cesarean delivery exceeds 1,000 ml. Intrahepatic cholestasis of pregnancies was based on: (1) onset of generalized pruritus in the second or third trimester of pregnancy; (2) bile acid level >10 μmol/L; and (3) spontaneous relief within 3 weeks after delivery. The standard of diagnosing polyhydramnios by B-ultrasound is amniotic fluid index (AFI) ≥25 cm or amniotic fluid volume (AFV) ≥8 cm. Placental abruption is defined as after 20 weeks of gestation or during childbirth, the placenta in its normal position is partially or completely removed from the wall of the uterus before the fetus is delivered.

Low birth weight was defined as birthweight below 2,500 g at delivery. Growth discordance was defined as a birthweight difference greater than 25% between twins. Neonatal asphyxia was classified according to *Hjalmarson (1981)*. Transient tachypnea, respiratory distress, and other types of respiratory disturbances were included in the analysis. Stillbirth is the death of a fetus in the uterine cavity after 20 weeks of gestation. Malformation was mainly defined whether the defecti was fatal or potentially life-threatening or likely to lead to serious handicap or major cosmetic defect if not surgically corrected. As to the rate of stillbirth and malformation, the numerator is the number of stillbirth and malformation, while the denominator is the number of twin deliveries. NICU admission and birth weight discordance (BWD) >25% calculated by the following formula: ((birth weight of larger twin–birth weight of smaller twin)/ birth weight of larger twin) ×100%.

Statistical analysis was performed using the Statistical Package for the Social Sciences, version 22.0 (SPSS Inc., Chicago, IL, USA). Differences between groups were tested statistically using the chi-square test for categorical data and the independent-sample *t*-test for continuous variables. A multiple logistic regression model was established to assess

**Table 1  Maternal characteristics and delivery modes in twin pregnancy.**

|  | SC ($n = 221$) | IVF ($n = 249$) | *P* value |
|---|---|---|---|
| Age (y) | 28.44 ± 4.27 | 30.66 ± 3.80 | <0.001 |
| Primiparity | 138 (62.4) | 212 (85.1) | <0.001 |
| Gestational age (w) | 36.41 ± 1.65 | 36.07 ± 1.71 | 0.026[a] |
| Cesarean section | 206 (93.2) | 239 (96.0) | 0.081 |
| First Cesarean section[b] | 24/82 (29.3) | 10/36 (27.8) | 0.527 |
| Length of hospitalization (d) | 6.98 ± 4.92 | 7.34 ± 4.99 | 0.042 |
| Birth weight (kg) | 2490.57 ± 432.37 | 2484.05 ± 428.03 | 0.817 |

**Notes.**
Values are cases (%) or mean ± standard deviation.
[a] The difference in mean gestational length attributes to the difference in gestational length counting methods for the two groups.
[b] The comparison is restricted to women with at least one previous delivery.

the association between the selected variables and the chance of pregnancy complications. $P < 0.05$ was considered statistically significant.

## RESULTS

The maternal age and the proportion of primiparity in the IVF-conceived group were significantly higher than in the SC group (30.66 ± 3.80 versus 28.44 ± 4.27 years old, $P < 0.05$, and 85.5% versus 62.4%, $P < 0.05$, respectively). The average duration of hospitalization in IVF-conceived population was significantly longer than in SC women (7.34 ± 4.99 versus 6.98 ± 4.92 days, $P < 0.05$). In contrast, the proportion of cesarean section and were similar (96% versus 93.2%, $P > 0.05$) in both groups. When the comparison of first cesarean section between the two groups was restricted to women with at least one previous delivery, the proportion of first cesarean section were similar (27.8% versus 29.3%, $P > 0.05$). The average gestational age in the SC group was 36.41 ± 1.65 weeks, which was significantly longer than 36.07 ± 1.71 weeks in the IVF-conceived group (Table 1). The birth weight between the two groups were similar (2490.57 ± 432.37 kg versus 2,484.05 ± 428.03 kg, $P > 0.05$).

After adjustment for maternal age and primiparity, regression analysis revealed there were no significant differences between the two groups ($P > 0.05$) in terms of maternal complications, with one exception of anemia ($P = 0.05$), as described in Table 2.

Neonatal outcomes are shown in Table 3. There were no significant differences in terms of the incidence of low birth weight, growth discordance, neonatal asphyxia and admission for NICU ($P > 0.05$). The stillbirth and malformation were described in Table 4; two cases in two different twins were included in SC group. Specifically, one is intrauterine fetal death infant, and the other is cystic mass of chest wall infant. In the IVF group there were four stillbirths and three malformed infants, including four intrauterine fetal death infants, two congenital heart malformation infants and one congenital high jejunal atresia infant. Moreover, the seven cases came from seven different pairs. However, statistical analysis was not performed on stillbirth and malformation due to the small number and rare conditions.

**Table 2  Maternal complications.**

| Complications | SC (*n* = 221) | IVF (*n* = 249) | Adjusted OR | Adjusted 95% CI | *P* value |
|---|---|---|---|---|---|
| Gestational diabetes mellitus | 32 (14.5) | 56 (22.5) | 1.24 | 0.74∼2.10 | 0.42 |
| Hypertensive disorder | 27 (12.2) | 29 (11.6) | 0.95 | 0.52∼1.77 | 0.88 |
| Premature rupture of membranes | 23 (10.4) | 25 (10.0) | 0.72 | 0.37∼1.38 | 0.32 |
| Maternal anemia | 110 (49.8) | 141 (56.6) | 1.51 | 1.01∼2.26 | 0.05 |
| Placenta previa | 5 (2.3) | 12 (4.8) | 2.08 | 0.66∼6.61 | 0.21 |
| Placental accreta | 7 (3.2) | 13 (5.2) | 1.33 | 0.48∼3.69 | 0.58 |
| Postpartum hemorrhage | 16 (7.2) | 23 (9.2) | 1.20 | 0.58∼2.48 | 0.63 |
| Intrahepatic cholestasis of pregnancies | 11 (5) | 14 (5.6) | 1.25 | 0.51∼3.09 | 0.63 |
| Polyhydramnios | 8 (3.6) | 9 (3.6) | 0.80 | 0.28∼2.31 | 0.68 |

Notes.
OR is the abbreviation of odds ratio, and CI is the abbreviation of confidence interval.

**Table 3  Neonatal outcomes.**

| | SC (*n* = 221) | IVF (*n* = 248) | Adjusted OR | Adjusted 95% CI | *P* value |
|---|---|---|---|---|---|
| Growth discordance | 15 (6.8) | 14 (5.7) | 0.76 | 0.34∼1.70 | 0.50 |
| Low birth weight | 139 (62.9) | 142 (57.3) | 0.97 | 0.65∼1.44 | 0.86 |
| Neonatal asphyxia | 16 (7.2) | 14 (5.7) | 0.90 | 0.58∼1.42 | 0.66 |
| NICU admission | 54 (24.4) | 64 (25.8) | 0.80 | 0.36∼1.76 | 0.57 |

Notes.
Values are cases (%) or mean (standard deviation); OR is the abbreviation of odds ratio, and CI is the abbreviation of confidence interval.

**Table 4  Stillbirth and malformation among the two groups.**

| SC(n) | IVF(n) |
|---|---|
| Intrauterine fetal death (1) | Intrauterine fetal death (4) |
| Cystic mass of chest wall (1) | Congenital heart malformation (2) |
| | Congenital high jejunal atresia (1) |

# DISCUSSION

In the past few years, growing interests has been paid to the role of IVF-ET treatments in obstetric and neonatal outcomes of twin pregnancies, but the results remain inconsistent (*Geisler et al., 2014*; *Helmerhorst et al., 2004*; *McDonald et al., 2005*; *Moini et al., 2012*; *Pourali et al., 2016*; *Qin et al., 2015*; *Sun et al., 2016*; *Vasario et al., 2010*). The current study compared the obstetric complications and neonatal outcomes of dizygotic twin gestations conceived via IVF with those from SC twin pregnancies.

During the past two decades, the rate of cesarean section in China mainland has risen rapidly. Elective and emergency caesarean section may contribute to this situation (*Geisler et al., 2014*). Moreover, there is no exception for the twin pregnancy population. Consistent with the study by *Moini et al. (2012)*, the rate of caesarean deliveries in the present study was similarly high in both twin pregnancy groups. Twin pregnancy has been known to

be associated with increased morbidity and mortality for the mother and neonate (*Rao, Sairam & Shehata, 2004*). This intrinsic risk may be responsible for the high prevalence of cesarean section in this study. In contrast, some studies have shown similar, but lower rates of caesarean delivery. Lambalk et al. reported a much lower rate of caesarean delivery (30% and 37%) in both twin gestations groups (*Lambalk & Van Hooff, 2001*). However, several other studies have demonstrated higher rates of caesarean delivery among IVF-conceived twin pregnancies (*Domingues et al., 2014*; *Geisler et al., 2014*; *Helmerhorst et al., 2004*; *Pourali et al., 2016*; *Vasario et al., 2010*). These results indicate that differences in study population and methodology for twin pregnancies related studies likely lead to the conflicting results.

In this study, the gestational age in the IVF-conceived group was significantly shorter than that in the SC group. However, the gap between the two groups was as small as a difference of 0.34 weeks, which means a little less than two and a half days. Meanwhile, 2~5 days' in vitro culture time before embryo transfer should also be added into the patients' pregnancy duration of IVF-conceived group. Accordingly, the difference between the two groups may be meaningless though it is statistically significant.

Actually, the difference in gestational time is the indication of the incidence of prematurity. Significantly shorter gestational time in IVF-conceived population has been shown in similar studies done by *Nassar et al. (2003)*, *Kallen et al. (2010)* and *Caserta et al. (2014)*. On one hand, this condition may be attributable to the infertility history and the importance of childbirth in the IVF group, obstetricians hence prefer to perform cesarean section ahead of schedule (*Pourali et al., 2016*; *Vannuccini et al., 2016*). On the other hand, families facing infertility usually believe that a trial of vaginal birth is more risky than cesarean section. Therefore, pregnant women are more inclined to choose cesarean section earlier than the expected date of childbirth.

Several previous studies have demonstrated that, in contrast to natural singleton pregnancy, IVF-conceived singleton gestation is independently associated with increased obstetric complications (*Farhi et al., 2013*; *Pinborg et al., 2013*). But does this effect also occurs in twin pregnancies when compared with SC twin pregnancies? The findings are conflicting. Some studies have shown that IVF treatments in dizygotic twin gestations are related to increased risks of obstetric complications, specifically preeclampsia, intrahepatic cholestasis of pregnancies (ICP) (*Sun et al., 2016*), and GDM (*Moini et al., 2012*; *Pourali et al., 2016*). In contrast, this relationship was not observed in several other studies (*Anbazhagan et al., 2014*; *Geisler et al., 2014*; *Moini et al., 2012*; *Vasario et al., 2010*). In addition, a systematic review demonstrated that perinatal mortality in twins conceived via ART was 40% lower compared with SC twins (*Helmerhorst et al., 2004*). When comparing the obstetric outcomes of IVF-conceived dichorionic twin pregnancies with those of SC dizygotic twin pregnancies, this study showed similar incidences of maternal and perinatal complications in terms of GDM, hypertensive disorder, premature rupture of membranes, maternal anemia, placenta previa, placental accrete, postpartum hemorrhage, drug use, ICP, and polyhydramnios. The only exception was maternal anemia, which was more frequently observed in the IVF twins group. However, the marginal $P$ value ($p = 0.05$) is inadequate to allow valid estimation of the real prevalence of maternal anemia in the

present study population, and the exact conclusion warrants further assessment based on a larger sample size.

It has been suggested that some infertility factors and IVF treatment itself may lead to the increased risks of perinatal complications. Unfortunately, because of the inadequate information obtained during the primary study, we could not address the effects of different infertility factors on perinatal outcomes. Thereby, our study still leaves open the question of whether one or a combination of infertility factors results in adverse perinatal outcomes.

With regard to the neonatal outcomes of twin pregnancies from IVF and natural conception, the literature provides contradictory findings. One meta-analysis (*Hansen et al., 2013*) and one well-designed study (*Qin et al., 2015*) concluded that multiple pregnancies resulting from IVF-ET were at significantly greater risks for adverse pregnancy outcomes than SC multiple pregnancies. Similarly, *Saccone et al. (2017)* performed a study including 668 women and showed that IVF-conceived oligohydramnios pregnancies had a higher rate of spontaneous preterm birth than SC dichorionic pregnancies. It is hypothesized that underlying infertility factors and invasive IVF-ET manipulations contribute to this condition (*Hansen et al., 2013*; *Nassar et al., 2003*; *Saccone et al., 2017*). However, in the current study, IVF treatments did not show any adverse effects on neonatal outcomes. The conclusion was in agreement with that in several previous studies, which also have failed to find the increased risks when comparing the neonatal outcomes according to mode of conception (*Anbazhagan et al., 2014*; *Pourali et al., 2016*; *Sun et al., 2016*; *Vasario et al., 2010*). Differences in study population and methodology may account for the conflicting data.

Although the present study used strict inclusion criteria, limitations do exist. First, some patients conceived by ovulation induction and artificial insemination may have been categorized into the SC group, and it has been reported that negative perinatal outcomes are more frequently observed in multiple pregnancies resulting from induction and artificial insemination compared with those conceived naturally (*Qin et al., 2015*). This may underestimate the relationship between IVF-ET and adverse outcomes. Second, the retrospective nature of data collection and the single-center study may restrict the strength and the quality of evidence. A well-controlled, multicenter study may be more convincing to address the issue. In addition, low number of cases may also lead to insufficient power of this kind of study.

## CONCLUSION

Although older age and shorter gestational age were more frequent in the IVF-conceived twin pregnancy population, this study did not show any increased risks for pregnancy-related complications and adverse neonatal outcomes in dizygotic twin pregnancies following IVF. With these fundamental data, this study could provide a reference for perinatal care and clinical ART treatment and help to inform infertile parents about the potential risks of IVF.

## ACKNOWLEDGEMENTS

We would like thank all personnel of the Reproductive Center, Department of OB & GYN, the 2nd Affiliated Hospital & Yuying Children's Hospital of Wenzhou Medical University, for their assistance in this work.

### Funding

This study was supported by grants from the Zhejiang Provincial Natural Science Foundation (LQ15H040006) and the Wenzhou Municipal Science and Technology Fund (Y20140615 and Y20160008) of China. The funders had no role in study design, data collection and analysis, decision to publish, or preparation of the manuscript.

### Grant Disclosures

The following grant information was disclosed by the authors:
Zhejiang Provincial Natural Science Foundation: LQ15H040006.
Wenzhou Municipal Science and Technology Fund: Y20140615, Y20160008.

### Competing Interests

The authors report no declarations of interest, and the authors alone are responsible for the content and writing of the paper.

### Author Contributions

- Hua Chen conceived and designed the experiments, performed the experiments, analyzed the data, contributed reagents/materials/analysis tools, prepared figures and/or tables, authored or reviewed drafts of the paper, approved the final draft.
- Ying Wan conceived and designed the experiments, performed the experiments, analyzed the data, contributed reagents/materials/analysis tools, prepared figures and/or tables.
- Haitao Xi, Jing Cheng and Chunfang Zhu analyzed the data.
- Weijue Su contributed reagents/materials/analysis tools, authored or reviewed drafts of the paper.
- Jieqiang Lv conceived and designed the experiments.
- Xinmei Wu conceived and designed the experiments, analyzed the data, approved the final draft.
- Junzhao Zhao conceived and designed the experiments, contributed reagents/materials/analysis tools, authored or reviewed drafts of the paper, approved the final draft.

### Data Availability

The raw measurements and raw data are available in the Supplemental File.

### Supplemental Information

Supplemental information for this article can be found online at http://dx.doi.org/10.7717/peerj.6638#supplemental-information.

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
