# Peer review of "Obstetric and perinatal outcomes of dizygotic twin pregnancies resulting from in vitro fertilization versus spontaneous conception: a retrospective study"

_PeerJ, doi:10.7717/peerj.6638_

## Round 0.1 · original submission · Major Revisions

Dear Authors,
The Reviewers found your manuscript very interesting, moreover recommending a thorough revision in order to achieve publication.
I would suggest to take into consideration the Reviewers' comments, discuss and incorporate them within your manuscript in order to reach the standard requested for publication.

It is strongly recommended that you have a colleague who is proficient in English and familiar with the subject matter, or a professional editing service, review your manuscript.

Best regards

Salvatore Andrea Mastrolia
Peerj Academic Editor

Reviewer 1 ·

Basic reporting

.

Experimental design

.

Validity of the findings

.

Additional comments

In all, this study is well designed and has addressed an important question. However, I still have the following concerns.

1. The presentation of the abstract is in chaos. It should be organized logistically and some conjunction words could be used to make the construction clearer. For example: The present study aims to … We found that … In conclusion…

2. It will be easier to grasp the design of this study if a workflow can be provided. The data of childbirth, twins, fetal reduction, and so on can be presented here.

3. The diagnosis of dichorionic pregnancy is important in this study. Therefore, how the diagnosis was made should be described in detail.

4. Please define the complications clearly, including maternal and neonatal complications. For example, how is growth discordance diagnosed?

5. Please list the specific malformations identified in this study.

6. The English of this paper is poor. There are plenty of grammatical mistakes in this manuscript, especially in the abstract. Some mistakes are listed as follows.
1) To address whether the twin pregnancy modes, specifically conceived spontaneously or by In Vitro Fertilization (IVF), is associated with adverse obstetric and perinatal outcomes.
2) 470 dichorionic twin pregnancies were divided on the basis of conception mode: 249 resulted from IVF and 221 were conceived spontaneously.
3) A higher risk of stillbirth and malformation (4.8% versus 1.4%, P < 0.05) and increased neonatal intensive care unit (NICU) admissions (23.4% versus 12.3%, P < 0.05, respectively) were observed in the IVF-conceived group.
4) The subjects in this study covers Wenzhou and its surrounding area, South-East Zhejiang province, China.

·

Basic reporting

The submission is a self-contained study that adheres to PeerJ policies. Raw data are supplied. English editing by a native or professional speaker is advised to ensure that an international audience can clearly understand your text.

The manuscript includes sufficient introduction and background which demonstrate how this study fits into the broader field of knowledge. The relevant prior literature is appropriately referenced.

However, some corrections in presentation of the study aim should be made. The paragraph with the study aim should be rewritten. This paragraph should only comprise the current first two sentences describing the objective. The remaining two sentences (3 and 4) should be transferred to discussion, at the end of the first paragraph, as the explanation of study novelty.

Moreover, currently it is written that “This study is thus designed to assess perinatal and neonatal outcomes of dichorionic twin pregnancies”. However, authors did not mention that the actual study aim was to compare the outcome of naturally conceived twins with IVF conceived ones. Therefore, the aim should be rewritten (This study is thus designed to assess perinatal and neonatal outcomes of dichorionic twin pregnancies conceived naturally or by IVF).

Experimental design

This is an original research. Research question is well defined, relevant and meaningful. Investigation seems to be performed according to high technical and ethical standards. However, the study method is not well presented.

Authors state that a strict selection of examined women was performed. What were the inclusion and exclusion criteria for the study? Were only IFV twins included or did authors consider twins conceived with other forms of assisted reproduction (IUI, ICSI, ET)?

It is not clear what was regarded as gestational age? Did authors mean gestational age at birth? Furthermore, what was considered as “duration of hospitalization”? Was it measured only after delivery, or some women had to be hospitalized prior to delivery due to complications? Please write everything clearly.

Explain with more details the sentence „The medical diagnosis was categorized according to the criteria proposed by Sun et al“. Which diagnoses?

Which perinatal complications were recorded? Which antenatal maternal complications were followed? Authors presented numerous data in the tables which were not at all mentioned in the methods.

How was the outcome of twin pregnancies assessed? What was the main outcome variable in this study and what were the assessed confounding factors? One of the most important negative outcomes seems to be stillbirth as well as having pregnancy-related complications. Outcome measures should be explained in details. Moreover, all analyses should be related to this outcome in particular.

I do understand that most Journals have strict word limitations, but study design and methodology need to be described with sufficient information to be fully understandable and reproducible by other researchers.

Validity of the findings

Data is statistically sound and controlled. However, applied statistical analyses are basic. I would advise authors to perform some more robust analyses (i.e. regression analysis that could test for confounding factors which might influence twin pregnancy outcome, or at least ANOVA or some other analysis depending on study aim).

Tables 1 and 3 should present all p values between groups (just like the Table 2).

Discussion is adequately written.

Conclusions are linked to original research question and based on the obtained results.

Additional comments

This study deals with in interesting and important issue of perinatal and neonatal outcomes of dichorionic twin pregnancies regarding their conception method. Growing interests has been paid to the role of IVF-ET in obstetric and neonatal outcomes of twin pregnancies. Still, as the results of available literature remain to be inconsistent this study is current and relevant.

It is not advisable to start the sentence with the number. Authors should rewrite such sentences in the abstract. These sentences could start with: study included / there were / etc.

Reviewer 3 ·

Basic reporting

1. Let us start with Table 1. The fact that women who have IVF are older than expected and more often have no previous child is well-known and easily explained and lines 139-140 are hardly needed. The authors find a higher rate of “first cesarean section” after IVF than in the control group. If I understand it right can this measure be meaningful only if the woman has at least one previous delivery. If I count it right, there are in the control material 83 women with one or more previous delivery, among which 44 had C-section (53%) while after IVF there are 37 women with one or more previous delivery, among which there are 10 with C-section (27%). The higher rate in the Table for IVF pregnancies is thus only an expression of the higher rate of first pregnancies. It seems that the difference in the Table just mirrors the obvious fact that the rate of first deliveries in the IVF group is high. This line should be removed, I think.
2. The difference in mean gestational rate is given as 0.34 day and statistically significant. When expressed in hours, this means 8 hours. Is this difference clinically or biologically meaningful? More important is prematurity. In a large study (Kallen et al., BJOG 2010; 117: 676-682, not quoted by the authors) based on 1545 IVF and 8675 non-IVF dizygotic twins the adjusted odds ratio for <32 weeks was 1.52 (95% CI 1.18-1.97) with week 37-38 as a reference.
3. Table 2 shows data on maternal conditions. One reaches statistical significance, that is, gestational diabetes. The given p-value, I suppose, is not adjusted for the fact that ten different comparisons have been made. The probability to get one p-value of 0.026 among ten tests is 1-(1-0.026)10 which is not significant. It may anyway be true as an increased risk of various maternal complications has been described by at least some authors. To this is added that in the present manuscript no adjustment has been made for the fact that for instance age an parity differ between the two groups. I think, therefore, that the authors should state their finding with less emphasis.
4. Table 3. Here are some matters which should be clarified. As far as I understand, the percentages are counted on the number of twin births, not on the number of twins. This means that one is describing how large chance it is that one or both twins were stillborn or malformed, for instance. If, for instance, both twins were malformed, were they counted as two (and then the denominator should be number of twins, not number of women) or as one (and then the denominator should be number of twin deliveries). I think the latter is the best solution – because of the tie between the two twins, but anyway it has to be clarified how one has counted. Furthermore, I think one should divide the outcome in stillbirths and malformations as these outcomes are very different. Finally, with such low numbers as in this situation, chi-square should perhaps be replaced by Fischer exact test. Again no adjustments have been made for the different characteristics of the groups which may contribute to the observed differences.
5. Generally speaking it is more informative to present odds ratios with 95% CI than percentages and p-values, preferably odds ratios at least adjusted for maternal age and parity. Women undergoing IVF often differ also in other aspects from non-IVF women but I don’t know how relevant these differences are in China, e.g., smoking and BMI.
Minor comments
1. Line 104: Neonatal birth weight – birth weight is always ne0onatal. Change to birthweight. What is meant with “neonatal conversion rate” – do you mean NICU transfer rate?
2. In some places you write “elder age”, I think “higher age” or “older age” is to be preferred.
3. Table 2: Heading. What do you mean with periternal complications? Anyway, I think the listed complications all are maternal.
4. In the Introduction there are statements on a high percentage of twin births after IVF. This was universally true up till the last decade(s) then single-embryo transfer has become the common mode of IVF in some countries. I think this should be pointed out in the Introduction as the comparison between IVF and spontaneous twins is of less interest in that setting- I think the last sentence of the Introduction is a bit exaggerated, considering the relatively small size of the study and its weaknesses, pointed out above. Line 100: what strict selection?

Experimental design

The basic data appear to be all right but above I have pointed out some deficiencies in the analysis of them. The most obvious defect is that no consideration has been taken to confounding factors, notably maternal age and parity.

You found no effect on pregnancy diagnoses (except for gestational diabetes). Have you considered if the numbers are large enough to demonstrate moderate effects? This can be evaluated from odds ratios with confidence intervals.

Validity of the findings

See above. The finding on gestational diabetes may well be random, the effects on stillbirths and malformations should be better presented (preferably with exact malformation diagnoses so the reader can see what has been included).

Additional comments

The English is understandable. I have pointed out some errors which should be changed but an editing of an English-speaking person would be could useful.

It would have been very nice if you in greater detail discussed previous literature. Now you state that results vary and explain it by some general phrases. In what studies were all twins compared, not dizygotic twins? When were the studies large enough to have a power to demonstrate anomalies in pregnancy diagnoses or outcome?

---

## Round 0.2 · Major Revisions

Dear Authors,

The Reviewers found your manuscript very interesting.

While Reviewer #1 was more positive and generally in favor of publication, Reviewer #3 still raised concerns and suggested publication to be considered only after a major revision of the manuscript is achieved.

In light of the Reviewer's comments, I personally evaluated the manuscript and consider that the authors should carefully consider the issues raised by Reviewer #3.

I would so far suggest to answer all comments within a rebuttal letter, discuss and incorporate them in a revised version of your manuscript in order to reach the standard requested for publication.

With personal regards

Salvatore Andrea Mastrolia, MD
PeerJ Academic Editor

Reviewer 1 ·

Basic reporting

pass

Experimental design

pass

Validity of the findings

pass

Additional comments

No more comments

Reviewer 3 ·

Basic reporting

There are a number of sentences which have to be clarified as pointed out in the review.
Literature references OK.
Figure OK, Tables need corrections as pointed out in he review. Raw data in Excell.
Actual hypothesis not stated. All conclusions (except possibly on ICU transfer) uncertain because of low power due to small numbers.

Experimental design

The study is within the scope of the journal.
Research question stated in broad outlines.
No ethical problems but weaknesses in some technical aspects.
In the revised manuscript, most methods used are described but the statistical analysis has some weak points as no consideration to ties between the two infants in a twin pair is taken.

Validity of the findings

Mainly negative results but the relatively small data set makes an absens of a difference uncertain. Actually, there is only one finding of a difference between the two groups which is of some interest, ICU transfer. A detailed analysis of the reasons for this would be welcome.


Some data are not robust, e.g., gestational duration and the concept of malformations, see review!

Additional comments

The authors have made a number of changes in the manuscript and have commented on my referee comments. I think there are still a number of major and minor things to correct before the manuscript can be accepted.
Comments
1. I pointed out that two “findings”, the higher age and lower parity of women in the IVF group, are self-evident. In spite of this these are listed in the Abstract. I suggest that the sentence starting in line 34 is removed. I am afraid the Abstract will be very thin – the only convincing finding worth mentioning is the increased risk of NICU transfers.
2. The authors have not taken into consideration my point that a first CS is obviously the case when the delivery is the woman’s first delivery and that IVF women are characterized by having more first deliveries than SC women. A crude comparison as made by the authors of the frequency of first CS is therefore rather meaningless – when one compares the rate of first CS among women who had at least one previous delivery (and therefore could have had a previous CS), the opposite is found. In their answer the authors say that they have added something to the Discussion on this point, but I cannot find it. The statement that the IVF group has a higher rate of first CS remains (line 159). I suggest this line is removed.
3. It is now clear that pregnancy duration is determined differently for the two groups: 14 days before embryo transfer in the IVF group and based on LMP for SC women. Actually, the first estimate should take culture time into consideration (2-5 days usually) and the uncertainty of the second estimate is well known – so the small mean difference between IVF and SC is rather meaningless even if it is statistically significant. The Abstract statement of a shorter gestational duration should be removed and in the text the measured difference should be commented on.
4. After adjustment no difference between the two groups remains with respect to GDM. This is demonstrated in Table 2. I suggest that the column P-value is removed as it can only confuse the reader. In the heading of the Table, I think it is enough to have Adjusted OR, 95%CI and P value. The marginally significant effect on maternal anemia should be commented upon: probably a result of mass significance. – I suggest that you only give a maximum of two decimals. These comments are valid also for Table 3.
5. In Table 3 it is said that 194 SC had low birth weight and this is 44% which must mean that it is calculated on 442 individuals. This is all right but the statistical comparison with IVF will be complicated because of the strong tie between the weights of the two twins. Now it does not matter much for most variables, only admission to NICU differs between the groups. Also for this variable there may be a tie but the difference is so large that it can hardly be explained this way.
6. Table 4 defines stillbirths and malformations. These two outcomes are added in Table 3 in spite of the fact that they are rather different biologically. In the SC column there is actually only one stillbirth and one “cystic mass of chest wall” – if you want to include the third case, neonatal death, you should use the term perinatal death instead of stillbirth. In the list of IVF cases there are four stillbirths – were they in four different twin pairs of in fewer pairs? Again a tie between the twins may complicate analysis. There are two cases of malformations (one cardiac defect, one jejunal atresia). One is a case of papyraceous fetus – is this not a case of early fetal death which should result in exclusion from the study? One case of intraventricular hemorrhage is hardly a stillbirth or a malformation. What is meant with umbilical cord bloomen? I have never heard that expression. Is it a malformation? – The material is really too small to study rare conditions like these and if one want to compare unusual outcomes between the two groups, these must be defined a priori. I think one could just write in the text that one and four infants in the two groups were stillborn and that one and two cases, respectively, were malformed.
7. I have a number of minor comments:
Line 39 (Abstract): --- adjusting for maternal age and parity. In one edition of the Abstract it says promiparity (I suppose you mean primiparity), in the other Abstract parity. Which is it?
Line 79: why write monochorionic and dizygotic, why not use monozygotic and dizygotic or monochorionic and dichorionic?
Line 125: When cesarean delivery exceeds 1000ml. Re-formulate!
Line 129: - the water amount of sheep—What is meant? I suppose this is a definition of polyhydramniosis but I don’t understand what sheep do here.
Line 141: one ‘the’ is enough. ‘Number of twins’ is not quite clear – it could also be the number of twin deliveries. In this connection, point out the complication with a tie between data for the two infants in a twin pair.
Line 148: A ----model was established------, adjusting for potential---. I don’t think that data can be significant, differences can.
Line 165: But you have shown that the higher rate of GDM was due to age and parity! If you want to present the crude percentages you should add the explanation, now mentioned first on line 173.
Line 260: The infertility factors do not lead to insufficient power – low number of cases does!
Table 3: I think 15 growth discordances in 249 twin pairs are 6.0%. Actually, I don’t suppose that the papyraceus fetus can (and has) been included so it should be 15/248, still 6.0%.

---

## Round 0.3 · Major Revisions

Dear Authors,

I very much appreciated the efforts provided in addressing the Reviewer's comments. However, the reviewers still have some important criticisms which at the moment do not allow it to be accepted for publication in PeerJ.

Due to the interest of the subject, I have decided to give an additional round for comments but I strongly encourage the Authors to take into consideration the Reviewer's comments, discuss, and incorporate them within the manuscript in order to reach the standard requested for publication

Best regards

Salvatore Andrea Mastrolia
PeerJ Academic Editor

·

Basic reporting

1. Basic Reporting

The submission adheres to PeerJ policies. Raw data are supplied.

The manuscript includes sufficient introduction and background which demonstrate how this study fits into the broader field of knowledge. The relevant prior literature is appropriately referenced.

All necessary corrections were performed.

However, authors have rewritten the abstract according to one of the reviewers advise. On the other hand, by doing so they have made it less informative. I advise that instead of current abstract authors write:

The study aimed to investigate if twin pregnancy conception modes (spontaneously or by In Vitro Fertilization - IVF) are associated with adverse obstetric and perinatal outcomes. Sample included 470 dichorionic twin pregnancies out of which 249 resulted from IVF and 221 were conceived spontaneously. Women in the IVF-conceived group apart from being older and more often nulliparous had all other investigated parameters similar to the counterparts in the spontaneous conception group. A higher risk of stillbirth and malformation (4.8% versus 1.4%, P< 0.05) and increased neonatal intensive care unit (NICU) admissions (23.4% versus 12.3%, P < 0.05, respectively) were observed in the IVF-conceived group. After adjusting maternal age and primiparity, the results showed that there were no significant differences between the two groups (P > 0.05) in terms of maternal antenatal complications and neonatal outcomes. In conclusion, IVF treatment does not increase risks for pregnancy-related complications and adverse neonatal outcomes in dizygotic twin pregnancies.

Experimental design

This is original research. Research question is well defined, relevant and meaningful. The investigation seems to be performed according to high technical and ethical standards.

All necessary corrections were performed.

Validity of the findings

Data is statistically sound and controlled.

The discussion part is adequately written.

Conclusions are linked to the original research question and based on the obtained results.

All necessary corrections were performed.

Additional comments

This study deals with an interesting and important issue of perinatal and neonatal outcomes of dichorionic twin pregnancies regarding their conception method. Growing interest has been paid to the role of IVF-ET in obstetric and neonatal outcomes of twin pregnancies. Still, as the results of available literature remain to be inconsistent this study is current and relevant.

All necessary corrections were performed.

Reviewer 3 ·

Basic reporting

The English may benefit from some polishing but is understandable.
Literature references rather selected but adequate.
Article structure OK
As apparent from my detailed comments there are still unclear parts.

Experimental design

No comment, see omments to authors

Validity of the findings

Some questions about data interpretation remain.

Additional comments

The authors have made a number of changes in their manuscript. I still have some unanswered questions and comments.
Major comments
1. Table 1 marks four significances: high maternal age and primiparity (which is a truism), mean gestational age, and “first cesarean section”. As I pointed out, the difference in mean gestational length is completely explained by the difference in gestational length determination for the two groups. It amounts to 0.34 weeks which means a little less than two and a half days. As now pointed out in the text (Line 205 but it should say 0.34 weeks, not days) this is reasonably an effect of the fact that culture time (2-5 days) was not included in the pregnancy length estimates for IVF twins. If this cannot be adjusted in the analysis, I suggest that a note is placed in Table 1 explaining the apparently significant difference.
2. The second finding in Table 1 is an increased risk for First Cesarean Section. In my previous review I pointed out that this is a direct effect of the fact that there are more primiparous mothers in the IFV group than in the SCB group – if a woman has her first delivery and C-section, then it must be her first C-section! One therefore has to restrict the comparison to women with at least one previous delivery (and then this finding disappears).
3. Table 4 in the previous manuscript has been removed as far as I can see but is still mentioned in line 174. According to this text and Table 3 there should be three malformed or stillborn infants in the SC group among them (according to Table 4 in the previous edition) one is a neonatal death. If one wants to add this into the analysis one should call it perinatal death and not stillbirth. In the IVF group there were four stillbirths and three malformed infants. The latter may (judge from the previous Table 4) be two heart defects and one jejunal atresia. But in Table 3, there are three and 12 cases listed and I think it should be three and seven, not twelve. Then – if we accept the debatable method to add the two very different outcomes perinatal deaths and malformations – we have 3/221 and 7/247 which (using exact Fisher analysis) gives a p value of 0.37 and an OR of 2.1 with a 95% CI of 0.5-12.7 thus far from statistically significant and with very low precision in the estimate. One gets roughly the same values if on uses the number of twins instead of number of twin births in the denominators. The mark of statistical significance in Table 3 thus seems incorrect. In Line 174 it is said that NICU admission is not statistically significantly increased – but in Table 3 it is marked as significant. Actually, as far as I can understand, this is the only convincing difference between the two groups. The odds ratio for NICU transfer of at least one of the twins is 2.2 (95% CI 1.3-3.7). Another matter is what can be the explanation to this.
4. I cannot anywhere in the manuscript find any comment on the fact that there are ties between data for the two twins in any twin pair why statistical analyses get complicated, notably in relatively small materials. Let us take the example of stillbirths. There were four stillbirths in the IVF material – were they in four different pairs or were two of them in the same pair? Obviously the risk for one of the twins to be stillborn is increased if the other is stillborn.
5. I also think that it would be useful if the authors point out that some of the findings of no difference between SC and IVF twins could be due to the limited material, notably in some of the outcomes listed in Table 2, e.g., placenta previa. The odds ratio for this is 2.2 (95% CI 0.7-8.0) – the study thus does not exclude an eight time increased risk! So be careful to say that your study shows that there is no increased risk, as stated in the Abstract. What you can say is that your study did not reveal an increased risk.
Minor comment
1. Generally you have given authors names in Italian style in the manuscript but in the first paragraph of the Introduction it is not.
2. Line 95: In the IVF-conceived---, line 108: in the first trimester, line 126 exposed part--, line 160 Remove And.

---

## Round 0.4 · Minor Revisions

Dear Authors,

the Reviewers are favorable to the publication of your manuscript in PeerJ after a minor revision.

Please incorporate or discuss the suggested changes and submit a revised version of your manuscript in order to achieve publication .

Best regards

Salvatore Andrea Mastrolia
PeerJ Academic Editor

·

Basic reporting

Adequate. No more comments.

Experimental design

Adequate. No more comments.

Validity of the findings

Adequate. No more comments.

Additional comments

Authors did not correct the Abstract as suggested. It is still short and uninformative. In the previous revision I have written the whole abstract as it is supposed to be in the text. The only thing that authors had to do was to copy and paste it in their text. But, they did not do that. Authors need to correct the abstract as suggested.

Reviewer 3 ·

Basic reporting

See previous reviews

Experimental design

See previous review

Validity of the findings

What is still missing is that the authors have not (as far as I can find) pointed out the relatively small size of the material which makes the "negative" outcome of the study less valuable. This will, however, be obvious for every reader of the paper.

Additional comments

The majority of my concerns have been answered and the text has been adequately modified.

---

## Round 0.5 · accepted · Accept

Dear Authors,

I would like to compliment with you for the efforts provided in addressing the Reviewers' comments.

Your manuscript has been considered suitable for publication and can be accepted in its current form.

Best regards

Salvatore Andrea Mastrolia
PeerJ Academic Editor